

# Optimized virtual reality design through user immersion level detection with novel feature fusion and explainable artificial intelligence

Ali Raza[1], Amjad Rehman[2], Rukhshanda Sehar[3], Faten S. Alamri[4], Sarah Alotaibi[5], Bayan Al Ghofaily[2] and Tanzila Saba[2]

[1] Department of Software Engineering, University of Lahore, Lahore, Pakistan
[2] Artificial Intelligence & Data Analytics Lab CCIS, Prince Sultan University, Riyadh, Saudi Arabia
[3] Institute of Computer Science, Khwaja Fareed University of Engineering and Information Technology, Rahim Yar Khan, Pakistan
[4] Department of Mathematical Sciences, College of Science, Princess Nourah Bint Abdulrahman University, Riyadh, Saudi Arabia
[5] Department of Computer Science, College of Computer and Information Sciences, King Saud University, Riyadh, Saudi Arabia

Corresponding author
Faten S. Alamri, fsalamri@pnu.edu.sa

## ABSTRACT

Virtual reality (VR) and immersive technology have emerged as powerful tools with numerous applications. VR technology creates a computer-generated simulation that immerses users in a virtual environment, providing a highly realistic and interactive experience. This technology finds applications in various fields, including gaming, healthcare, education, architecture, and training simulations. Understanding user immersion levels in VR is crucial and challenging for optimizing the design of VR applications. Immersion refers to the extent to which users feel absorbed and engrossed in the virtual environment. This research primarily aims to detect user immersion levels in VR using an efficient machine-learning model. We utilized a benchmark dataset based on user experiences in VR environments to conduct our experiments. Advanced deep and machine learning approaches are applied in comparison. We proposed a novel technique called Polynomial Random Forest (PRF) for feature generation mechanisms. The proposed PRF approach extracts polynomial and class prediction probability features to generate a new feature set. Extensive research experiments show that random forest outperformed state-of-the-art approaches, achieving a high immersion level detection rate of 98%, using the proposed PRF technique. We applied hyperparameter optimization and cross-validation approaches to validate the performance scores. Additionally, we utilized explainable artificial intelligence (XAI) to interpret the reasoning behind the decisions made by the proposed model for user immersion level detection in VR. Our research has the potential to revolutionize user immersion level detection in VR, enhancing the design process.

# INTRODUCTION

Virtual reality (VR) and immersive technology have revolutionized the way we expose and interact with digital environments (*Riches et al., 2023*; *Jerald, 2015*). Virtual reality involves a simulated three-dimensional environment created by computers, which users can navigate and engage with. VR is typically based on wearing a head-mounted display that immerses the user in a virtual world, often accompanied by other sensory inputs such as touch and sound. This technology strives to establish an immersive experience, making the user perceive themselves as being physically located within the virtual setting (*Creed et al., 2023*; *Nasralla, 2021*). Immersive technology, on the other hand, encompasses a vast range of technologies that aim to fully engage the user's senses and create a highly immersive experience. This includes virtual and augmented reality (AR), mixed reality (MR), and other related technologies (*Lombard et al., 2015*). AR overlays digital content onto the real world, while MR blends virtual and real elements seamlessly. These immersive technologies have found applications in numerous fields, including gaming, entertainment, healthcare, education, and training (*Fernandes et al., 2023*; *Rather et al., 2023*), offering exciting possibilities for creating realistic and engaging experiences that were once only limited to our imagination.

VR applications have gained significant attention and popularity in various fields due to their numerous advantages (*Ning et al., 2023*). VR technology provides an immersive and interactive experience by simulating a realistic three-dimensional environment, enabling users to interact with digital objects and scenarios in real time. One of the critical advantages of VR applications is their ability to enhance training and education. In domains like healthcare, aerospace, and defense, VR enables learners to undergo training for intricate tasks and situations within a secure and manageable setting, thereby diminishing the likelihood of mistakes and mishaps (*Creed et al., 2023*). Moreover, VR applications have proven valuable in architectural design and urban planning, as they enable architects and urban planners to visualize and explore their designs before the actual construction begins. This leads to better decision-making, improved collaboration, and cost savings. Another advantage of VR applications lies in their potential for therapeutic purposes (*Chen, Wu & Huang, 2023*). Virtual reality can treat anxiety disorders, phobias, and post-traumatic stress by exposing patients to virtual environments that gradually desensitize them to their fears.

User immersion levels in VR environments play a crucial role in enhancing the design of virtual reality experiences (*Valluripally et al., 2023*). Immersion refers to the degree to which users feel fully engrossed and connected to the virtual world they are experiencing (*Slater & Wilbur, 1997*). By understanding and leveraging user immersion levels, designers can create VR environments that provide more captivating and realistic experiences. When users are highly immersed, their cognitive and emotional engagement increases, leading to a heightened sense of presence and an enhanced feeling of being physically present within the virtual environment (*Elor & Kurniawan, 2020*). This heightened immersion can create a more memorable and impactful VR experience for users. Designers can utilize realistic graphics, interactive elements, spatial audio, and haptic feedback to create a sense

of presence and increase immersion levels. By considering user immersion as a critical factor in VR design, designers can create more compelling and engaging virtual reality experiences that effectively transport users to new, immersive digital worlds.

Artificial intelligence techniques have been increasingly employed to detect and assess user immersion levels in VR environments (_Hong & Ge, 2022_). By leveraging machine learning algorithms, researchers have sought to capture and analyze various user behavioural and physiological signals to gauge the extent of immersion experienced by individuals within the VR setting (_Wu & Han, 2023_). These techniques encompass a range of approaches, including computer vision, natural language processing, and sensor data analysis. Computer vision algorithms enable the tracking and analyzing of user movements, facial expressions, and eye gaze patterns, providing valuable insights into the user's engagement and presence in the virtual environment (_Ahuja et al., 2023_). Integrating artificial intelligence techniques holds great promise for advancing our understanding of user experiences in VR environments and informing the design of immersive virtual experiences that captivate and engage users to a greater degree.

The primary research contributions related to virtual reality are followed as:

- A novel Polynomial Random Forest (PRF) method is proposed, which extracts polynomial and class prediction probability features from the virtual reality experience benchmark dataset, generating a new feature set.
- Advanced deep and machine learning methods are employed for comparisons. K-fold validation, hyperparameters optimization, and computational complexity analysis are conducted to validate the performance.
- An XAI approach is employed to interpret the reasoning behind the decisions made by the proposed model for detecting the user's immersion level in VR.

The remaining research study is categorized as 'Literature analysis' comparatively analyzed the literature for virtual reality applications. Our proposed study methodology is demonstrated in 'Proposed methodology'. 'Results and discussion' is based on the results of applied artificial intelligence approaches. Our study findings are concluded in 'Conclusions and future work'.

## LITERATURE ANALYSIS

Literature analysis in VR involves conducting a comprehensive examination of existing research studies. Through a systematic review of the available literature, we can better understand the current state of knowledge, identify research gaps, and uncover valuable insights. The scope of literature analysis on VR applications is broad and encompasses various topics such as gaming, education, healthcare, architecture, and training simulations. The literature summary is provided in Table 1.

In _Suhaimi, Mountstephens & Teo (2022)_, the authors proposed a virtual reality environment aimed at inducing four classes of emotions: happy, sad, terrified, and bored, for the purpose of classifying human emotions. The dataset used in this study comprised a total of 32 subjects, including seven females and 25 males. The VR video dataset was sourced from participants on YouTube and offline recordings, and brain signals were

**Table 1  The analyzed literature summary analysis.**

| Ref. | Year | Dataset | Proposed technique | Performance accuracy |
|---|---|---|---|---|
| *Suhaimi, Mountstephens & Teo (2022)* | 2022 | VR videos data | ML | 85% |
| *Cha & Im (2023)* | 2023 | FeMG data | VR and fEMGbased metaverse | 86% |
| *Xiao (2023)* | 2023 | FER2013 | GAN Network | 72% |
| *Cha & Im (2022)* | 2021 | FeMG data | LDA | 89% |
| *Pei et al. (2023)* | 2023 | VR videos | RBFNN | 91% |
| *Mateos-García et al. (2023)* | 2023 | Stress recognition in automobile drivers | ML | 90% |
| *Uyanık et al. (2022)* | 2022 | VREED | ML | 76% |
| *Geraets et al. (2021)* | 2021 | VR image and videos | VR Task | 75% |
| *Bulagang, Mountstephens & Teo (2021)* | 2023 | Empatica E4 wearable wristband, VR headset | ML | 80% |
| *Zheng, Mountstephens & Teo (2020)* | 2020 | VR headset and Eye tracking data | ML | 70% |

collected using an EEG handset. Various classical machine learning models were applied in this study. The research results indicate that the proposed support vector machine model achieved an accuracy score of 85%, which is considered low.

The research proposed by *Cha & Im (2023)* applies covariate shift adaptation approaches in the feature and classifier domains to build an fEMG-based FER system that is resistant to electrode shifts. This system is designed for use in VR-based metaverse applications (*Saba et al., 2019*), enabling reliable facial expression identification even when the electrode placements shift. In this experiment, the fEMG dataset is used, which records eleven expressions from different subjects. The study results demonstrate that without using the suggested method, the classification accuracy decreased from 88% to 79% when the electrode placements were changed. However, when the suggested covariate shift adaptation method was employed, the accuracy dramatically increased to 86%. Although this improvement is significant, it remains relatively low compared to the baseline accuracy.

In *Xiao (2023)*, the authors proposed a unique feature fusion technique for facial recognition in VR. This approach was developed because the conventional method's face expression detection algorithm (*Meethongjan et al., 2013*) performs poorly in real-world scenarios. In the experimental study, the authors utilized the FER2013 dataset and extracted convolutional features from VGG16 and ResNet50 networks. These features were then combined with HOG features and classified using support vector machine (SVU) (*Sharif et al., 2019*). The initial classification accuracy achieved by SVM was 67%. To further enhance the accuracy, the authors employed a GAN network and achieved an accuracy of 72%. However, it is important to note that the performance scores of this study are relatively low.

*Cha & Im (2022)* proposed to create a new technique for enhancing facial expression recognition (FER) performance by combining labelled datasets from other users. In this

study, experiments were conducted using systems for social VR applications based on the fEMG dataset to classify eleven different facial expressions. The results of the study indicate that an accuracy of 89.4% could potentially be achieved by applying the linear discriminant analysis (LDA) adaptation technique. It is important to mention that the level of accuracy achieved does not surpass that found in existing studies.

This research was proposed by *Pei et al. (2023)* to investigate how EEG signals can be used to distinguish between different emotional states in VR scenarios and to enhance the computational speed and accuracy of emotional valence identification. In this experiment, emotional VR videos, which are largely typical, were used to collect subjects' emotions. Radial basis function neural networks (RBFNN) and DBF were employed to classify the model accuracy. The study results indicate that the proposed RBFNN achieved the highest accuracy of 91%, which is considered low.

*Mateos-García et al. (2023)* proposed research to create a system for identifying stress during various driving circumstances in a car by combining the usage of sensors and the opportunities provided by VR. In this experiment, a total of 8 subjects (four male and four female) aged between 24 to 50 years were recruited. The dataset used in this study is titled "Stress Recognition in Automobile Drivers" and was produced in 2007. It is freely accessible on the PhysioNet platform. Different advanced machine learning models were applied to determine the accuracy, and the model that achieved the highest accuracy was the Decision Tree (DT), which achieved a 90% accuracy rate. However, further improvements are still needed.

*Uyanık et al. (2022)* proposed an improved method for EEG-based emotion detection using the recently released VREED dataset, which is publicly available. The experiment involved 15 male and female subjects who were exposed to a total of 60 VR videos, each lasting 4 s, belonging to three different categories. All videos were shot in 3D by the Shanghai Film Academy. Various machine learning models were employed to achieve the highest accuracy using differential entropy (DE) features in different classifiers. The model that yielded the highest accuracy was SVM, which achieved a result of 76.22% with a computation time of 2.06 s. However, these results are relatively low and demonstrate a high error rate.

This work was proposed by *Geraets et al. (2021)* was to measure and develop the ability to recognize emotions. VR permits the administration of dynamic, realistic stimuli within a social situation. In this study, twenty avatars were dispersed throughout the virtual street area. Each avatar displayed emotions, including anger, disgust, fear, happiness, sorrow, surprise, or neutrality, for 10 s whenever a participant moved within a two-meter range of the avatar. The VR activity involved rating the emotions of virtual characters (avatars) in a VR street scene and recording eye-tracking in VR. After this experiment, VR achieved a total recognition accuracy of 75%, matching that of the photo and video task.

*Bulagang, Mountstephens & Teo (2021)* aimed to test if heart rate (HR) signals can be utilized in a virtual reality (VR) environment to classify four different moods using Russell's emotion model. In this experiment, a total of 20 subjects (12 male and eight female), ranging in age from 20 to 28 years, participated. The Emotiv E4 wristband sensor was employed to record heart rate data, while a VR headset was used to present videos

to the subjects. According to the experimental findings, the accuracy of intra-subject emotion classification across the four classes ranged from 45.4% to 100%. These findings demonstrate the promising potential of HR as an approach for emotion classification in a four-class setting using VR, particularly in predicting emotions for participant-specific data. Intra-subject classification yielded accuracy rates of over 80% using SVM and KNN algorithms.

*Zheng, Mountstephens & Teo (2020)* proposed a method for recognizing emotions in a virtual environment using pupillometry. The experiment involved using the Pupil Labs app to collect eye-tracking data. As part of the data collection process, the participants' eyes were calibrated. Initially, all participants wore a VR headset with an eye-tracking attachment. Pupil Capture was utilized to record the students' data. The experiment stimuli consisted of videos presented in 360 degrees while wearing the VR headset. The pupil diameter was measured using an eye tracker and used as the sole criterion for emotion classification. Three advanced machine-learning techniques were employed for emotion classification. Among all the models, SVM achieved the highest average accuracy of 57%. However, the accuracy for properly identifying emotions from the LA/NV quadrant was relatively poor, around 70%.

The presence of user immersion in VR can be determined in various ways (*Gougeh et al., 2022*). VR headsets like the HTC Vive, Oculus Rift, and PlayStation are employed to quantify immersion levels. The duration of time users spend in a virtual environment, as well as the occurrence of motion sickness, serves as a self-assessment of their susceptibility to motion sickness while engaging in VR (*Kim, 2020*). Higher ratings on the scale, which spans from 1 to 10, suggest a higher likelihood of experiencing motion sickness.

## Research gap and challenges

The identified research gap through literature analysis is followed as:

- Mainly, classical machine learning approaches were used in the literature, which resulted in low-performance scores. The highest achieved accuracy score was 91%, indicating the need for advancements in methods to address this literature gap.

In the realm of image and video analysis within VR applications, current methodologies frequently suffer from suboptimal performance, primarily due to two significant factors. First, the datasets employed in training and testing these models predominantly consist of image and video data, which inherently contain various types of noise such as lighting variations, background clutter, and camera motion—that degrade the data quality. Second, many existing approaches have relied on classical feature engineering techniques, which fail to capture the high-level, complex patterns needed for advanced VR applications. This underscores the need for ensemble learning strategies, which integrate multiple learning algorithms to produce more accurate results. Leveraging ensemble learning for richer, deeper feature extraction can significantly enhance performance through a more nuanced understanding and utilization of the physical features of persons interacting within VR settings.

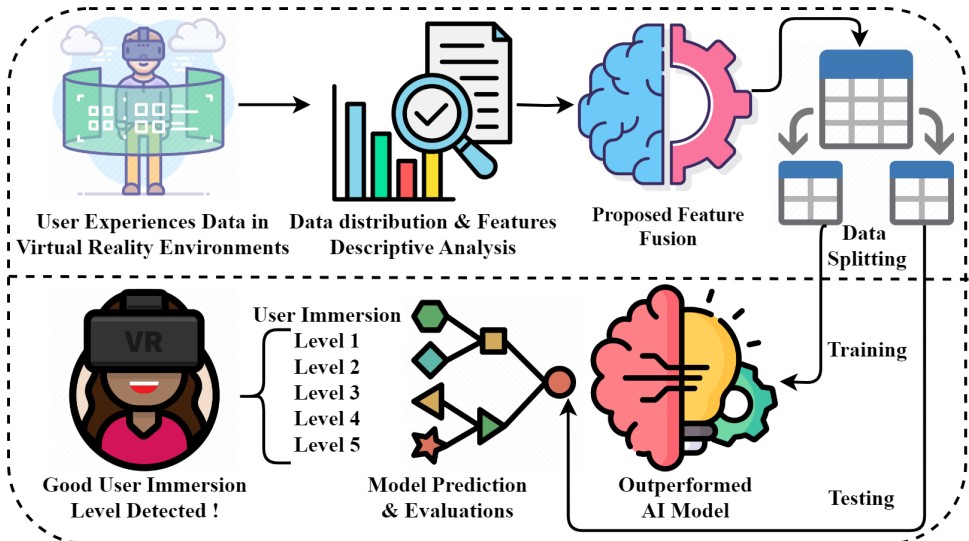

**Figure 1** **Novel proposed methodology architecture analysis.** Image credits: Exploratory Analysis free icon (pojok d: https://www.flaticon.com/free-icon/exploratory-analysis_12489889, Flaticon license); Brainstorm free icon (kmg design: https://www.flaticon.com/free-icon/brainstorm_3981729, Flaticon license); Machine learning free icon (Freepik: https://www.flaticon.com/free-icon/machine-learning_8637099, Flaticon license); Classification free icon (geotatah: https://www.flaticon.com/free-icon/classification_992257, Flaticon license); Vr Game free icon (Flat Icons: https://www.flaticon.com/free-icon/vr-game_3098910, Flaticon license); Avatar, female avatar, virtual reality icon (Evoria Studio: https://www.iconfinder.com/icons/3172990/avatar_female_avatar_virtual_reality_vr_vr_glasses_vr_headset_icon, Basic license); Split, table, tables icon (Axialis: https://www.iconfinder.com/icons/2306058/split_table_tables_two_icon, Basic license).

# PROPOSED METHODOLOGY

This section analyzes our proposed research methodology for detecting user immersion levels in virtual reality environments. The step-wise flow of the proposed study is illustrated in Fig. 1. In our study, we collected virtual reality data based on user experiences to conduct experiments. We performed a descriptive and distributional analysis of the data, focusing on feature states. Using a novel PRF method, we extracted polynomial and class prediction probability features from the virtual reality experience data, resulting in a new feature set. This newly generated feature set is then partitioned into training and testing sets in an 80:20 ratio. We employed advanced deep and machine-learning techniques for the newly created data. The performance scores of the superior AI model were evaluated using unseen testing data. Finally, an efficient AI model was developed to detect user immersion levels and enhance the design of virtual reality environments.

## Virtual reality experiences data

The virtual reality benchmark dataset (*Joshi, 2023*) based on user experience is used in our study to conduct experiments. The dataset collection involved 1,000 subjects. This dataset contains values that are derived from people's physiological reactions and feelings while using VR applications. This benchmark dataset includes information about the users'

**Table 2  The descriptive benchmark dataset features analysis.**

| Feature name | Data type | Description |
|---|---|---|
| User ID | int 64 | Each user participating in the VR experience has a specific identifier represented by this feature. To distinguish the data of each user in the dataset, it assigns them a unique ID. |
| Age | int 64 | This feature represents the age of the subjects who participated in VR as an integer value. |
| Gender | object | This feature identifies the gender of the subject, distinguishing between male and female. |
| VRHeadset | Object | This feature allows us to determine the type of VR headset the user uses during their VR experience. It could be a VR headset such as the HTC Vive, Oculus Rift, PlayStation VR, or any other type. |
| Duration | float 64 | This feature indicates the amount of time subjects spend in a virtual environment, measured in minutes. |
| Motion sickness | Int 64 | This feature reflects the user's self-assessment of their susceptibility to motion sickness while engaging in virtual reality (VR). Higher ratings on the scale, which ranges from 1 to 10, indicate a greater risk of experiencing motion sickness. |
| Immersion level | Int 64 | This targeting feature captures the extent to which users felt immersed in the VR experience. It demonstrates users' engagement and involvement with the VR experience. The user's level of immersion in the virtual environment is measured on a scale of 1 to 5, with 5 representing total immersion. |

preferences and dislikes, as well as details about their heart rate, skin conductance, and moods during virtual reality applications. Table 2 presents the results of the descriptive analysis of the dataset. Additionally, we conducted a data distribution analysis, as depicted in Fig. 2.

## Novel proposed approach

The working architecture flow of our novel PRF proposed approach is analyzed in this section. The feature generation mechanism is illustrated in Fig. 3. To begin with, the virtual reality data based on user experience is input to a polynomial function for extracting polynomial features (*Sukhbaatar et al., 2023*), and to a random forest approach for extracting class prediction probability features (*Raza et al., 2022b*). Then, the features extracted from both approaches are fused together to generate a new feature set. Finally, the newly generated feature set is used to conduct the proposed study experiments.

### *Polynomial features*

The polynomial features are designed to expand the feature space of input data by generating polynomial and interaction features. It creates a new feature matrix where each feature is transformed into all polynomial combinations up to a specified degree. This transformation allows the model to capture non-linear interactions between features, potentially leading to more accurate predictions.

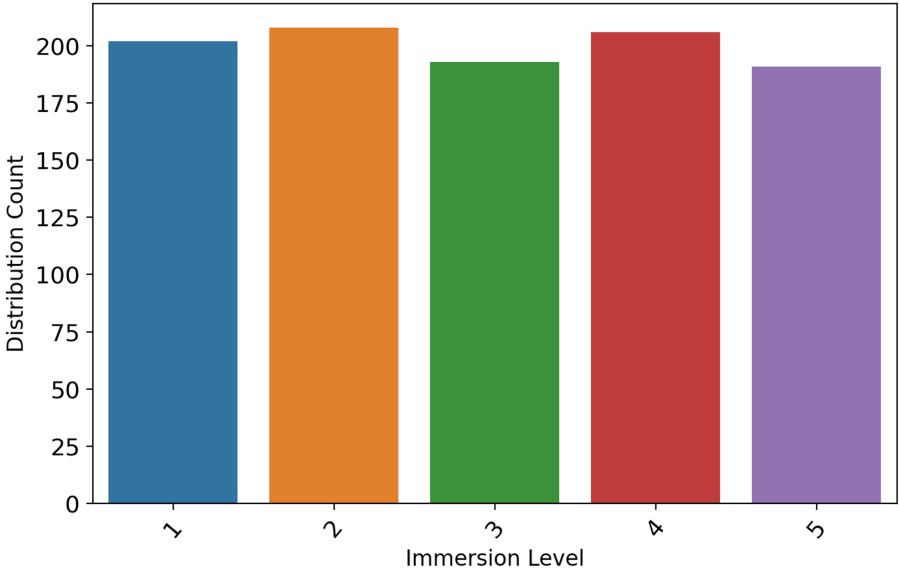

**Figure 2** The bar chart-based user immersion level data distributions analysis.

### Class prediction probability features

The class prediction probability features compute the probabilities based on the average outputs from all the trees within the random forest model. Specifically, for each tree in the forest, the probability assigned to a particular class for a given sample is calculated as the proportion of training samples of the same class that fall within the same leaf as the input sample. The random forest provides a robust estimation of class probabilities by averaging these individual probabilities across all trees, which is helpful for more accurate predictions

Algorithm 1 outlines the sequential process of the suggested method for combining features.

---

**Algorithm 1** PRF Algorithm

---

**Input:** Virtual reality data based on user experience.

**Output:** Novel fused features for detecting user immersion levels.

initiate;

1- $F_{pol} \longleftarrow PoL_{polynomial\ features}(Vd)$    // here $Vd$ belong to the Virtual reality data and $F_{pol}$ are predicted features.

2- $F_{rf} \longleftarrow RF_{probability\ features}(Vd)$    // here $Vd$ belong to the Virtual reality data and $F_{rf}$ are predicted features.

3- $NF_{features} \longleftarrow \sum\{F_{pol} + F_{rf}\}$    // $NF_{features} \in$ Novel fused features set used for detecting user immersion levels.

end;

---

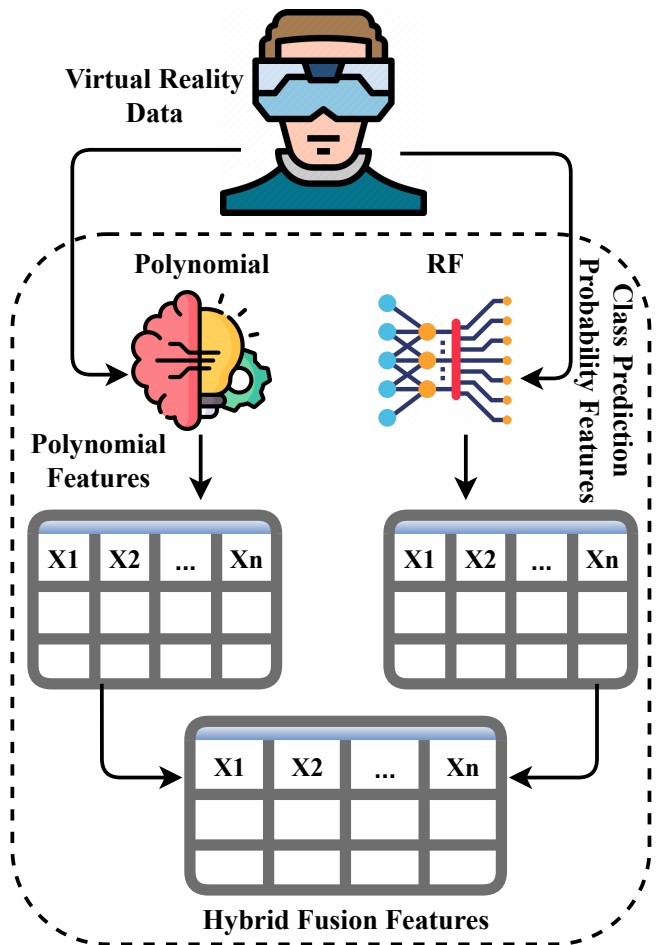

**Figure 3** **The architecture analysis of novel proposed feature fusion approach.** Image credits: Deep Learning free icon (Becris: https://www.flaticon.com/free-icon/deep-learning_2103832?related_id=2103718&origin=search, Flaticon license); Machine Learning free icon (Freepik: https://www.flaticon.com/free-icon/machine-learning_8637099, Flaticon license); Virtual Reality Glasses free icon (SBTS2018: https://www.flaticon.com/free-icon/virtual-reality-glasses_2199589, Flaticon license).

## Applied artificial intelligence techniques

AI is increasingly being used to enhance VR environment design (*Wu & Han, 2023*; *Raza, Munir & Almutairi, 2022*). One of the most promising areas of AI research in VR is the use of machine learning (*Kokash et al., 2024; Alam, Ahmed & Kokash, 2023*; *Rashid et al., 2021*) to detect user immersion levels. This information can then be used to improve the VR experience in several ways. AI-based user immersion level detection can also be used to improve the design of VR environments. By tracking user attention and engagement, AI can identify which elements of a VR environment are most effective.

For the detection of user immersion levels, we have applied state-of-the-art methods in comparisons, as described in Table 3. In addition, the layered architecture stack of the applied deep learning model GRU is analyzed in Table 4.

**Table 3  The descriptive working analysis of applied deep and machine learning approaches.**

| Applied technique | Description |
|---|---|
| Random forest (RF) | Random Forest (RF) is a collective learning technique that builds numerous decision trees during the training phase (*Raza et al., 2023a*; *Maher et al., 2023*). The output of the RF is the target class selected by most trees or the mean or average prediction of the individual trees. |
| Logistic regression (LR) | LR is a statistical model that predicts the probability values of a binary outcome (*León et al., 2023*). It is a type of regression analysis that uses a logistic function to approach the probability of a certain event occurring. |
| Support vector machine (SVM) | SVM is a supervised learning algorithm that can be used for classification and regression tasks (*Asbee et al., 2023*). SVMs work by finding the best fit hyperplane that separates two classes of data points. |
| Gated recurrent unit (GRU) | GRU is a kind of recurrent neural network (RNN) (*Raza et al., 2022a*) that utilizes gating mechanisms to control the flow of data information in and out of the network. GRUs have two gating mechanisms, the reset and update gates. |

**Table 4  The layer architecture stack analysis of applied deep GRU model.**

| Layer name | Output shape | Param # |
|---|---|---|
| gru (GRU) | (None, 64) | 12864 |
| dense_4 (Dense) | (None, 32) | 2080 |
| dropout_2 (Dropout) | (None, 32) | 0 |
| dense_5 (Dense) | (None, 5) | 165 |
| Total params | 15,109 | |

**Table 5  The hyperparameter setting analysis of applied deep and machine learning techniques.**

| Technique | Hyperparameters values |
|---|---|
| RF | n_estimators=100, max_depth=100, random_state=0 |
| LR | random_state=0, max_iter=100 |
| SVM | random_state=0, max_iter=500 |
| GRU | loss = 'categorical_crossentropy', optimizer = 'adam', activation='softmax' |

## Hyperparameter setting

Hyperparameters are parameters that control the learning process of applied AI techniques (*Raza et al., 2023b*). Typically established prior to initiating the training process, the values of these parameters can significantly influence the performance of the artificial intelligence model. The optimal values for hyperparameters can vary depending on the AI technique and the dataset being used. Therefore, we tuned the hyperparameters of applied AI methods to achieve high performance, as shown in Table 5.

## Performance evaluation criteria

In our research study, we employed a comprehensive set of performance evaluation criteria to assess our proposed model's effectiveness and robustness thoroughly. The primary metrics used are the accuracy score, precision score, recall score, and F1 score:

- The accuracy score gauged the overall correctness of the model across all classes.
- In contrast, the precision score assessed the model's ability to label as positive a sample that is indeed positive.
- The recall score measured the model's capacity to identify all relevant instances within a dataset.
- The F1 score offered a harmonic mean of precision and recall, balancing both metrics in scenarios where an equilibrium is crucial.

Furthering our evaluation, we utilized confusion matrix analysis, which helped visualize the algorithm's performance by detailing the number of correct and incorrect predictions concerning each class. K-fold cross-validation is another critical method applied, ensuring the model's reliability and stability by testing it across multiple subsets of the data to avoid overfitting and to provide a generalized performance estimate. Lastly, we incorporated XAI analysis in our evaluation to ensure transparency and interpretability of the model's decisions.

# RESULTS AND DISCUSSION

The results and discussion of our research on machine learning-based enhancing virtual reality environment design through user immersion level detection are presented in this section. To validate the effectiveness of our proposed research approach, we conducted a series of experiments using a diverse set of virtual reality scenarios.

## Development environment

In this study, we conducted all our experiments using the Python programming language, specifically version 3.0. The modules utilized to evaluate results included sklearn, matplotlib, seaborn, keras, and tensorflow. To carry out our experiments, we employed the Google Colab environment, an online platform that supports GPU backend and offers 90 GB of storage space along with 13 GB of RAM. The performance metrics utilized in our evaluation comprised accuracy score, precision score, recall score, and F1 score.

## Results with original features

The performance results of applied machine learning techniques with original data features are analyzed in Table 6. The accuracy, precision, recall, and f1 score values are determined in this analysis. The results show that when using the original features, the Random Forest (RF) technique achieved comparatively poor performance scores. On the other hand, the logistic regression (LR) and SVM techniques achieved higher scores compared to the RF technique, although not the highest. Additionally, the classification report indicates that the applied methods using original features achieved low-performance scores. Therefore, the analysis concludes that there is a need to enhance the results performance of the applied

**Table 6  Performance comparisons analysis of applied techniques with original features.**

| Technique | Accuracy | Target class | Precision | Recall | F1 |
|---|---|---|---|---|---|
| RF | 0.16 | 1 | 0.14 | 0.14 | 0.14 |
| | | 2 | 0.13 | 0.18 | 0.15 |
| | | 3 | 0.22 | 0.20 | 0.21 |
| | | 4 | 0.14 | 0.12 | 0.13 |
| | | 5 | 0.20 | 0.17 | 0.18 |
| | | Average | 0.17 | 0.16 | 0.16 |
| LR | 0.20 | 1 | 0.24 | 0.41 | 0.30 |
| | | 2 | 0.19 | 0.28 | 0.23 |
| | | 3 | 0.10 | 0.11 | 0.11 |
| | | 4 | 0.25 | 0.21 | 0.23 |
| | | 5 | 0.00 | 0.00 | 0.00 |
| | | Average | 0.16 | 0.20 | 0.17 |
| SVM | 0.23 | 1 | 0.50 | 0.03 | 0.05 |
| | | 2 | 0.19 | 0.13 | 0.15 |
| | | 3 | 0.00 | 0.00 | 0.00 |
| | | 4 | 0.26 | 0.73 | 0.38 |
| | | 5 | 0.12 | 0.10 | 0.11 |
| | | Average | 0.21 | 0.20 | 0.14 |

methods by employing advanced mechanisms, as the original data features yielded poor performance scores.

## Results with PCA features

For fair comparisons, we applied PCA and selected the three most important features to evaluate the results. PCA allows for dimensionality reduction, which can enhance the performance of machine learning algorithms. The performance analysis of different machine learning approaches using PCA features is summarized in Table 7. The analysis demonstrates that the LR algorithm showed poor performance compared to the RF and SVM methods. Using the PCA features, the RF method achieved a higher accuracy score of 0.23. The classification report indicates that results are improved compared to the original features when using PCA. This analysis wrap-up that the PCA approach achieves low performance scores, indicating a need for further performance enhancement.

## Results with novel features

Finally, we evaluated applied deep and machine learning approaches with novel features in this analysis. The performance of various machine learning methods was examined using a novel feature set, and the results are described in Table 8. The evaluation of the applied methods included measures such as recall, precision, and F1 score. The analysis revealed that the SVM method exhibited the lowest accuracy performance score of 0.26 when compared to other methods. On the other hand, the GRU and LR methods achieved good accuracy scores of 0.94 and 0.97, respectively. Surprisingly, the RF approach outperformed the others with a high accuracy score of 0.98 when using the novel feature set. The class-wise

**Table 7   Performance comparison analysis of applied approaches with PCA features.**

| Technique | Accuracy | Target class | Precision | Recall | F1 |
|-----------|----------|--------------|-----------|--------|------|
| RF | 0.23 | 1 | 0.23 | 0.27 | 0.25 |
|    |      | 2 | 0.21 | 0.23 | 0.22 |
|    |      | 3 | 0.22 | 0.26 | 0.24 |
|    |      | 4 | 0.26 | 0.25 | 0.26 |
|    |      | 5 | 0.18 | 0.12 | 0.14 |
|    |      | Average | 0.22 | 0.23 | 0.22 |
| LR | 0.20 | 1 | 0.27 | 0.35 | 0.31 |
|    |      | 2 | 0.16 | 0.26 | 0.20 |
|    |      | 3 | 0.17 | 0.20 | 0.18 |
|    |      | 4 | 0.21 | 0.21 | 0.21 |
|    |      | 5 | 0.00 | 0.00 | 0.00 |
|    |      | Average | 0.16 | 0.20 | 0.18 |
| SVM | 0.21 | 1 | 0.21 | 0.51 | 0.29 |
|     |      | 2 | 0.14 | 0.10 | 0.12 |
|     |      | 3 | 0.23 | 0.43 | 0.30 |
|     |      | 4 | 0.30 | 0.06 | 0.10 |
|     |      | 5 | 0.20 | 0.02 | 0.04 |
|     |      | Average | 0.22 | 0.23 | 0.17 |

performance report demonstrated that the proposed RF approach achieved a precision, recall, and F1 score of 0.99 for each. Based on this research analysis, we conclude that the novel feature set proposed in this study yields excellent results.

The time series results analysis during the training of the applied neural network approach, GRU, is illustrated in Fig. 4. The analysis demonstrates that during the first four epochs of training, low-performance scores are achieved with high-loss scores. However, after epoch 5, the GRU model extracted the optimal weights and enhanced the performance score. From epochs 7 to 8, the model accuracy ranged from 90% to 94%. This analysis wrap up that while the neural network technique achieved acceptable scores, it did not achieve the highest scores.

The confusion matrix results analysis of the applied method using novel features is demonstrated in Fig. 5. This analysis validates the performance of each applied method on unseen testing data. The analysis shows that the RF technique achieves the lowest error rate during classification, followed by LR and GRU. In comparison, the SVM technique achieved a higher error rate. Based on this research analysis, it can be summed up that the proposed RF techniques achieved the minimum error rates, thus validating their high-performance scores.

## Performance comparisons with all features

In the comparative analysis of machine learning techniques, RF, LR, SVM, and GRU are evaluated across three feature sets: original features, reduced via PCA, and newly proposed features. The performance was measured regarding classification accuracy, summarized in Table 9. Notably, the proposed features significantly enhanced model performance

**Table 8** Performance comparison analysis of applied methods with novel proposed features.

| Technique | Accuracy | Target class | Precision | Recall | F1 |
|---|---|---|---|---|---|
| RF | 0.98 | 1 | 1.00 | 1.00 | 1.00 |
| | | 2 | 1.00 | 0.95 | 0.97 |
| | | 3 | 0.97 | 1.00 | 0.99 |
| | | 4 | 0.98 | 0.98 | 0.98 |
| | | 5 | 0.98 | 1.00 | 0.99 |
| | | **Average** | **0.99** | **0.99** | **0.99** |
| LR | 0.97 | 1 | 1.00 | 1.00 | 1.00 |
| | | 2 | 1.00 | 0.92 | 0.96 |
| | | 3 | 0.92 | 1.00 | 0.96 |
| | | 4 | 0.98 | 0.98 | 0.98 |
| | | 5 | 0.98 | 0.98 | 0.98 |
| | | Average | 0.98 | 0.98 | 0.97 |
| SVM | 0.26 | 1 | 1.00 | 0.16 | 0.28 |
| | | 2 | 0.21 | 1.00 | 0.34 |
| | | 3 | 1.00 | 0.06 | 0.11 |
| | | 4 | 1.00 | 0.08 | 0.15 |
| | | 5 | 0.00 | 0.00 | 0.00 |
| | | Average | 0.64 | 0.26 | 0.18 |
| GRU | 0.94 | 1 | 0.88 | 0.97 | 0.92 |
| | | 2 | 0.97 | 0.92 | 0.95 |
| | | 3 | 0.97 | 1.00 | 0.99 |
| | | 4 | 0.94 | 0.98 | 0.96 |
| | | 5 | 0.97 | 0.85 | 0.91 |
| | | Average | 0.95 | 0.95 | 0.94 |

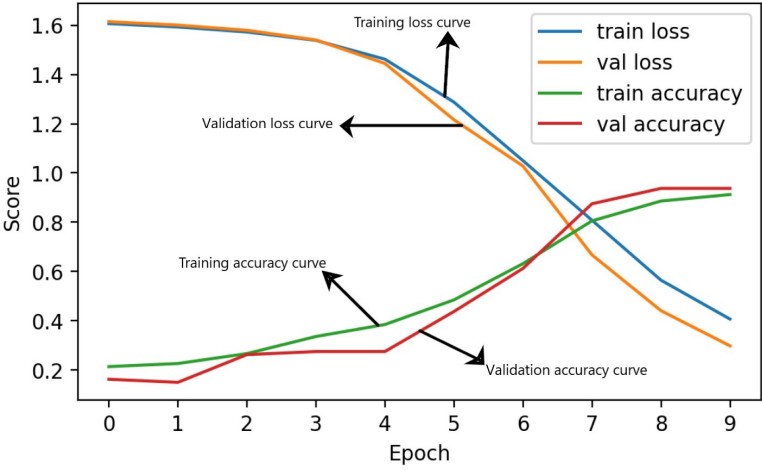

**Figure 4** The time series analysis of applied GRU model during training.

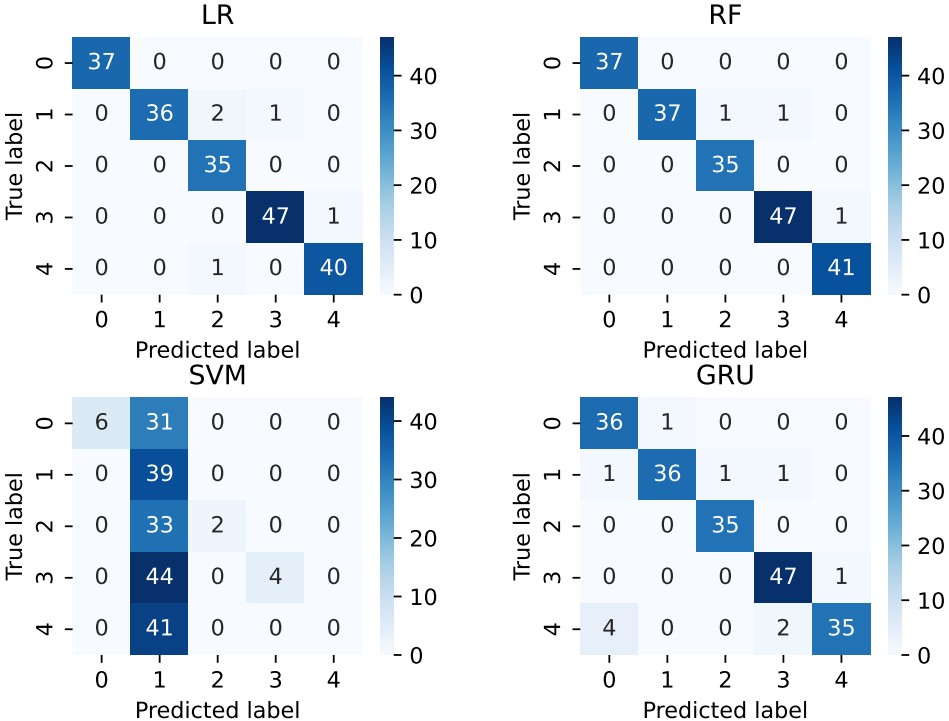

**Figure 5** The confusion matrix results analysis of applied techniques.

across nearly all techniques. For instance, RF's accuracy improved dramatically from 0.16 with original features to 0.98 with proposed features. A similar trend is observed in LR, where accuracy increased from 0.20 to 0.97. GRU also exhibited a substantial gain in accuracy from 0.22 with original features to 0.94 with the proposed features, indicating strong compatibility between the proposed features and sequence processing models. These results underline the critical impact of feature engineering on the efficacy of learning algorithms, particularly highlighting the potential of tailored feature sets to optimize model performance significantly.

## K-fold cross validations

The performance scores of each applied technique are validated using the k-fold cross-validation approach in this analysis. All used machine learning models underwent 10-fold cross-validation, as shown in Table 10. The k-fold analysis is evaluated using three different metrics: k-fold accuracy and standard deviation scores. The analysis illustrated that the LR and GRU techniques achieved a good k-fold accuracy score of 0.97. The analysis concludes that the RF technique outperformed others, with a high accuracy of 0.979 and a minimum standard deviation score of 0.01. The analysis shows that all applied methods are generalized for detecting user immersion levels in VR environments.

**Table 9  The performance comparisons with all features of applied learning methods.**

| Technique | Accuracy results with original features | Accuracy results with PCA features | Accuracy results with proposed features |
|---|---|---|---|
| RF | 0.16 | 0.23 | 0.98 |
| LR | 0.20 | 0.20 | 0.97 |
| SVM | 0.23 | 0.21 | 0.26 |
| GRU | 0.22 | 0.24 | 0.94 |

**Table 10  Performance validation results analysis of applied approaches with novel features.**

| Technique | K-Fold | K-Fold accuracy | (+/-) Standard deviation |
|---|---|---|---|
| RF | 10 | 0.979 | 0.01 |
| LR | 10 | 0.970 | 0.02 |
| SVM | 10 | 0.642 | 0.17 |
| GRU | 10 | 0.931 | 0.02 |

**Table 11  Computational complexity analysis of applied techniques.**

| Technique | Runtime computations (seconds) |
|---|---|
| RF | 0.68 |
| LR | 0.21 |
| SVM | 0.11 |
| GRU | 5.31 |

## Computational complexity analysis

The computational complexity results analysis of the applied techniques is examined in Table 11. The analysis is based on the runtime computations in seconds for each model. The analysis reveals that the lowest score achieved is 0.11 s; however, SVM also scored low-performance scores in comparison with others. The proposed RF approach achieved a computational complexity of 0.68 s with high-performance scores. The computational complexity of the proposed RF approach is low compared to the applied deep learning method in this analysis.

## Explainable AI analysis

XAI analysis (*Albahri et al., 2023*) has emerged as a critical area of research aimed at enhancing the transparency, interoperability, and decision-making processes of proposed AI models. One XAI analysis approach utilizes the SHAPE (Significance, Heterogeneity, Accuracy, Pragmatism, and Explanation) chart, as illustrated in Fig. 6.

The SHAPE chart in this analysis provides a structured framework for evaluating and understanding the strengths and limitations of our proposed AI model's explanations. The analysis demonstrates that the newly created features (f8, f10, f11, f7, and f9) have high relevance when predicting user immersion levels in VR environments. This analysis enhances the interoperability of our proposed RF technique during decision-making processes for unseen testing data.

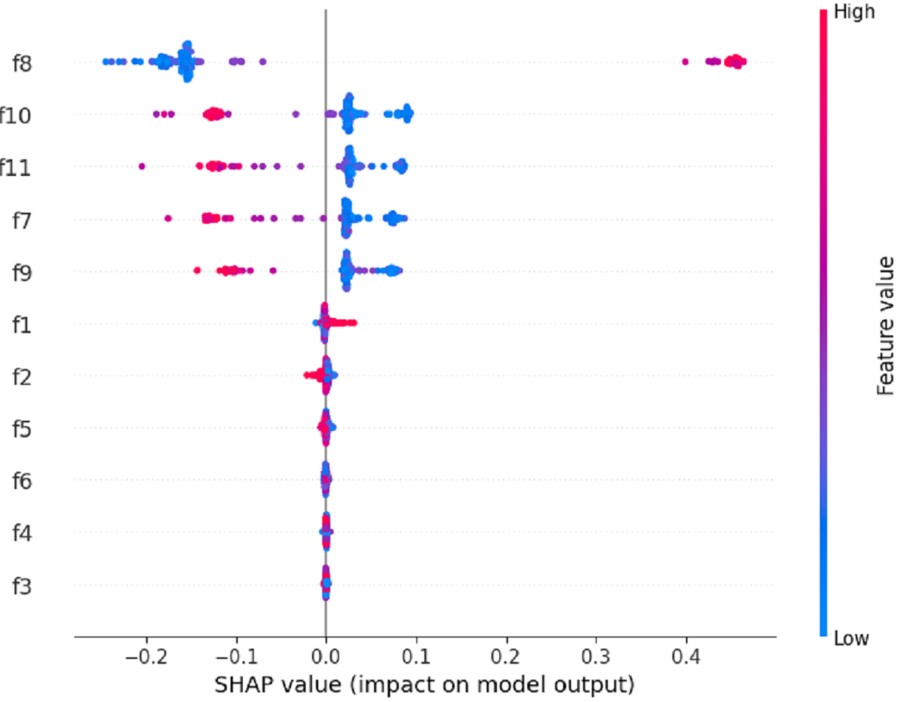

**Figure 6** **The SHAPE chart based XAI analysis of the proposed RF model.**

## State-of-the-art studies comparison

In our comparative analysis of state-of-the-art studies, we examined a range of machine-learning techniques to benchmark the effectiveness of our novel PRF method. As presented in Table 12, our ensemble learning approach, featuring the novel PRF method, demonstrated a superior performance accuracy of 98%. This outperforms the traditional machine learning techniques, such as SVM and LR, which have shown high accuracy in earlier studies. Notably, SVM, a frequently cited model due to its robustness in handling non-linear data, reached up to 93% accuracy, illustrating its effectiveness yet still falling short compared to our proposed method. Our PRF method leverages an ensemble of predictive models to fine-tune the accuracy and robustness, enhancing the detection of user immersion levels significantly.

## Discussions

Our proposed study aims to develop advanced deep and machine learning models for detecting user immersion levels in virtual reality (VR). To conduct our experimental analysis, we utilize a benchmark dataset consisting of user experiences in VR environments. The results of our study demonstrate that the use of original features yields low-performance scores. In order to enhance performance, we introduce a novel technique called PRF, which stands for feature engineering mechanisms.

The proposed PRF approach extracts polynomial and class prediction probability features to generate a new feature set. These newly generated features are then utilized for constructing the applied machine learning and deep learning models.

**Table 12 The state-of-the-art studies comparison analysis.**

| Ref. | Learning type | Proposed technique | Performance accuracy |
|------|---------------|--------------------|----------------------|
| *Lamb, Neumann & Linder (2022)* | Machine learning | SALEs | 85% |
| *Ke et al. (2023)* | Machine learning | Supprt Vector Machine | 93% |
| *Anwar et al. (2020)* | Machine learning | logistic regression | 86% |
| *Fathy et al. (2023)* | Machine learning | bagged trees algorithm | 71% |
| *Siyar et al. (2020)* | Machine learning | Supprt Vector Machine | 90% |
| **Ours** | **Ensemble learning** | **Novel PRF** | **98%** |

**Notes.**
The proposed model is shown in bold.

Numerous advanced deep and machine learning methods were employed for comparisons. Extensive research experiments showed that RF outperformed with a high performance accuracy of 98% compared to state-of-the-art studies. The K-fold cross-validation, hyperparameter optimization, and computational complexity analysis approaches were applied to validate the performance scores. Additionally, we utilized the XAI approach to interpret the reasoning behind the decisions made by the proposed model for detecting user immersion levels in VR.

# CONCLUSIONS AND FUTURE WORK

This study aims to detect user immersion levels in VR using an efficient AI model. We utilized data on user experiences in VR environments to conduct our experiments. We proposed a novel technique called PRF for feature engineering mechanisms. The PRF approach extracts polynomial and class prediction probability features to generate a new feature set. Advanced machine and deep learning methods were employed for comparisons. In-depth experimental investigations have demonstrated that the RF technique, leveraging the newly proposed PRF method, surpassed contemporary studies, attaining a remarkable performance metric of 98%. The K-fold validation, hyperparameters optimization, and computational complexity analysis approaches were applied to validate the performance scores. Additionally, we utilized the XAI approach to interpret the reasoning behind the decisions made by the proposed model for user immersion level detection in VR.

## Limitations and future work

In future work, we aim to minimize the computational cost of our proposed model. There is still a 2% error rate in performance accuracy, which can be further improved by applying more advanced learning methods. Additionally, we plan to collect and enhance the dataset from VR environments. We will also develop more advanced transfer learning-based approaches to enhance virtual reality design by detecting user immersion levels.

### Funding

This research is funded by Princess Nourah bint Abdulrahman University and Researchers Supporting Project number (PNURSP2024R346), Princess Nourah bint Abdulrahman University, Riyadh, Saudi Arabia. Prince Sultan University Riyadh Saudi Arabia supported Article Processing Charges (APC) of this publication. The funder provided full support regarding datasets, software, technology, infrastructure for study design, data collection and analysis, decision to publish, and preparation of the manuscript.

### Grant Disclosures

The following grant information was disclosed by the authors:
Princess Nourah bint Abdulrahman University, Riyadh, Saudi Arabia: PNURSP2024R346. Prince Sultan University Riyadh Saudi Arabia supported Article Processing Charges (APC) of this publication.

### Competing Interests

The authors declare there are no competing interests.

### Author Contributions

- Ali Raza conceived and designed the experiments, performed the experiments, prepared figures and/or tables, authored or reviewed drafts of the article, and approved the final draft.
- Amjad Rehman performed the experiments, performed the computation work, prepared figures and/or tables, and approved the final draft.
- Rukhshanda Sehar conceived and designed the experiments, analyzed the data, performed the computation work, authored or reviewed drafts of the article, and approved the final draft.
- Faten S. Alamri conceived and designed the experiments, performed the experiments, prepared figures and/or tables, authored or reviewed drafts of the article, and approved the final draft.
- Sarah Alotaibi performed the experiments, analyzed the data, prepared figures and/or tables, and approved the final draft.
- Bayan Al Ghofaily analyzed the data, performed the computation work, authored or reviewed drafts of the article, and approved the final draft.
- Tanzila Saba conceived and designed the experiments, performed the computation work, prepared figures and/or tables, authored or reviewed drafts of the article, and approved the final draft.

### Data Availability

The raw data are available at Kaggle: https://www.kaggle.com/datasets/aakashjoshi123/virtual-reality-experiences.

The code is available in the Supplemental Files.

## Supplemental Information

Supplemental information for this article can be found online at http://dx.doi.org/10.7717/peerj-cs.2150#supplemental-information.

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
