# Peer review of "Optimized virtual reality design through user immersion level detection with novel feature fusion and explainable artificial intelligence"

_PeerJ Computer Science, doi:10.7717/peerj-cs.2150_

## Round 0.1 · original submission · Major Revisions

Dear Authors,

Thank you for your submission, The experts in the field have reviewed the article with interest and they have pointed out that major considerations and improvements are needed to be incorporated in your paper to be considered further, I do agree with them and suggest to carefully consider the suggestions along with mine and resubmit.

Please write your rebuttal letter clearly stating the changes, their impact in the main manuscript and your responses.

As identified by the experts, The paper claims to propose a novel technique called Polynomial Random Forest (PRF) for detecting user immersion levels in VR. However, the novelty of the approach is questionable, as it primarily relies on combining existing machine learning techniques without introducing significant innovation in methodology or approach or what ?

While the paper discusses the potential implications of detecting user immersion levels in VR for optimizing VR application design, it fails to provide concrete examples or case studies demonstrating real-world applications or benefits. Without practical application scenarios, the relevance and impact of the research remain unclear and may not be useful for the readers.

The paper states that the proposed technique outperformed state-of-the-art studies with a high-performance score of 98%. However, the evaluation methodology and criteria for determining performance are not sufficiently detailed. Without a rigorous evaluation process and comparison with alternative methods, the validity and reliability of the reported results are called into question. Please clearly state this in your revision manuscript.

The paper focuses extensively on technical details such as machine learning algorithms, feature engineering mechanisms, and hyperparameter optimization, while neglecting to discuss the broader implications and potential limitations of the proposed approach. This narrow focus detracts from the paper's overall readability and relevance to a broader audience beyond the field of machine learning.

Reviewer 1 ·

Basic reporting

Here are some suggestions to improve the paper:
1. In the abstract of the manuscript, the authors used about one-third of the content to introduce the background, which is useless. Suggest the author quickly show the topic of the paper and point out the difficulties of detecting user involvement levels in VR.
2. In Section 2 (LITERATURE ANALYSIS), the authors only identified the research gap of the existing work at the end, but did not explain the reasons for the low-performance scores. As the contribution of the paper lies in designing new features, it is recommended that the authors analyze the reasons for the low performance of the existing methods and explain how this paper intends to address them.
3. In Section 3.2 (Novel proposed approach), feature extraction methods should be described in detail. Suggest providing a detailed explanation of the polynomial function and showing how the two kinds of features to be fused.

Experimental design

In this paper, the contribution lies in feature generation mechanisms, then the deep and machine learning approaches are used for classification based on the extracted feature. I have some suggestions to improve the experiment.

1. I think the comparison method in Tables 6-8 does not show the contribution of the paper well. It is recommended to compare the classification results of different features in a table.
2. In addition, the PRF proposed in the paper is a combination of two types of features. It is recommended to add a comparative experiment to compare the proposed features with the original two types of features.

Validity of the findings

no comment

Reviewer 2 ·

Basic reporting

This paper aims to detect user immersion levels in VR using an efficient machine-learning model. The authors utilized a benchmark dataset based on user experiences in VR environments to conduct our experiments. a novel technique called Polynomial Random Forest (PRF) was proposed for feature engineering mechanisms. Extensive research experiments show that random forest outperformed state-of-the-art studies, achieving a high-performance score of 98%, using the proposed PRF technique. The paper is well organized and written. However, there are some issues to be clarified.
1. How to evaluate immersion, apart from subjective feelings, is there an objective evaluation criteria.
2. Are there any similar research data sets in the world? The data set in this paper includes 1000 samples, which is a small number and whether it covers enough usage cases and scenarios.
3. It is recommended to add references to the method of comparison.
4. For the innovative method PRF proposed in this paper, it is suggested to add pictures to represent the innovation of the algorithm.

Experimental design

It is recommended to include more samples to enhance the generalization of the method.

Validity of the findings

The experiment is sufficient.

---

## Round 0.2 · accepted · Accept

Thank you for incorporating the comments and updating the paper, I am pleased to inform you about the acceptance of your manuscript . Thank you for your fine contribution.

Reviewer 1 ·

Basic reporting

The authors have well addressed the issues raised by the reviewer, and I think the current version could be accepted for publication.

Experimental design

No more suggestion

Validity of the findings

No more suggestion

Additional comments

No more suggestion